# A Review: Laser Welding of Dissimilar Materials (Al/Fe, Al/Ti, Al/Cu)—Methods and Techniques, Microstructure and Properties

**DOI:** 10.3390/ma15010122

**Published:** 2021-12-24

**Authors:** Sergey Kuryntsev

**Affiliations:** Department of Materials Science and Welding, Kazan National Research Technical University Named after A.N. Tupolev—KAI (KNRTU—KAI), 420111 Kazan, Russia; kuryntsev16@mail.ru; Tel.: +7-9872-901-953

**Keywords:** laser welding, dissimilar metals, weldability, microstructure, intermetallic layer, mechanical properties

## Abstract

Modern structural engineering is impossible without the use of materials and structures with high strength and low specific weight. This work carries out a quantitative and qualitative analysis of articles for 2016–2021 on the topic of welding of dissimilar alloys. It is found that laser welding is most widely used for such metal pairs as Al/Fe, Al/Ti, and Al/Cu. The paper analyzes the influence of the basic techniques, methods, and means of laser welding of Al/Fe, Al/Ti, and Al/Cu on the mechanical properties and thickness of the intermetallic compound (IMC). When welding the lap joint or spike T-joint configuration of Al/Fe, it is preferable to melt the steel, which will be heated or melted, by the laser beam, and through thermal conduction, it will heat the aluminum. When welding the butt-welded joint of Al/Fe, the most preferable is to melt the aluminum by the laser beam (150–160 MPa). When welding the butt-welded joint of Al/Ti, it is possible to obtain the minimum IMC and maximum mechanical properties by offsetting the laser beam to aluminum. Whereas when the laser beam is offset to a titanium alloy, the mechanical properties are 40–50% lower than when the laser beam is offset to an aluminum alloy. When lap welding the Al/Cu joint, under the impact of the laser beam on the aluminum, using defocusing or wobbling (oscillation) of a laser beam, it is possible to increase the contact area of electrical conductivity with the tensile shear strength of 95–128 MPa.

## 1. Introduction

The primary trend of modern structural engineering is the use of materials and structures with high strength and low specific gravity, which helps to reduce the weight of the final product [1,2,3,4]. An example of such materials is multi-materials or hybrid structures, consisting of several dissimilar metal alloys connected by welding, bolting, or brazing [5,6,7,8,9,10,11].

As a rule, the most processable and less labor-consuming connection type is welding, which provides the greatest strength and tightness. The weldability of dissimilar materials could be complicated by the difference in the physical properties of the materials being joined [5,6]. It requires the use of technologies based on thermal and mechanical impacts on the joined workpieces; among these are welding, brazing, and welding-brazing.

The body of a modern passenger car consists of approximately 84 kg of aluminum, 50 kg of steel, 9 kg of magnesium, and 10 kg of plastic by weight. Additionally, welded joints from dissimilar materials are widely used in the aircraft building, shipbuilding, carriage-building, aerospace, and electrical industries. Therefore, the issue of joining dissimilar materials is a promising and knowledge-intensive process task. An understanding of the physical, chemical and metallurgical processes occurring when joining dissimilar materials is the basis for choosing the type and method of welding, welding techniques and technology to obtain a joint with the required characteristics.

At fusion welding of dissimilar metals, the factors which affect the quality of the welded joints are:

Physical and thermophysical:Melting point;Boiling point;Thermal conductivity;Heat capacity;Coefficient of thermal expansion (CTE);Physics of material interaction with a heating source.

Chemico-metallurgical:
The ability to form solid solutions, chemical compounds, and eutectics;Difference in crystal lattices;Difference in the atomic sizes;Wettability of one metal by another (and vice versa);Electronegativity;Limiting solubility.

The main problem that reduces the mechanical and operational properties of dissimilar alloys welded joints is the formation of an IMC layer. In the case when the intermetallic phase is uniformly distributed in the metal volume between the grains and it is a dispersed reinforcing agent that inhibits dislocations or is a crystallization center with a high melting point [12,13]. The IMC role will be positive, and it will increase mechanical properties. However, if the IMC is present in the form of a continuous inclusion on the fusion line of two dissimilar metals, then, in this case, the transition line from the IMC to the metal will be a weak area due to the gradient and inhomogeneity of the microstructure, since the IMCs have high hardness and brittleness.

Almost all pairs of metals used in structural engineering form not only chemical compounds but also substitutional solid solutions of limited solubility, which depends on the temperature and interaction time. Table 1 shows that only copper and nickel have excellent weldability the remaining metal pairs generally have satisfactory or poor weldability.

The conditions of continuous solubility of two elements, stated by Hume-Rothery, are as follows [5,14]:(1)The elements must have crystal lattices of the same type;(2)The atomic diameters of the elements must differ by no more than 8–15%;(3)The elements should have similar electrochemical properties, which is observed when the electronic structure of their atoms is similar.

That is, when the nature of metals differs significantly, they do not fully dissolve among themselves, even in a molten state. In this case, it is difficult to obtain a monolithic high-quality welded joint by any type and method of welding. Based on the above, an important task is to ensure the uniformity of diffusion processes over the thickness of the butt-welded workpieces, which will make it possible to obtain a uniform IMC due to the precision of the laser beam impact.

Diffusion welding is the undisputed leader among the welding types that allow connecting significantly dissimilar materials, such as glass to metal, ceramics to glass, and others [15]. However, this welding type has some significant disadvantages compared to fusion welding, such as equipment complexity, low productivity, the limited size of workpieces, etc. Fusion welding is the most common technological process for joining metallic materials; 85–90% of all welded metal structures are obtained by fusion welding. The most common welding type is arc welding, however, over the last five years, laser welding has become more widespread due to the improvement of equipment, an increase in the efficiency of lasers, the multifunctionality of the laser beam, the massive application of technologies, and, as a result, due to the development and implementation of new production technologies [16].

The main fields of application for laser welding are:Welding of large thicknesses in one pass (up to 25–30 mm);Welding of small thicknesses (up to 1 mm);Welding of dissimilar metals.

When laser welding the dissimilar metals, the main advantage is a high welding speed (up to 833 mm/s) [17] and energy concentration (laser spot diameter up to 50 µm). It allows minimizing the interaction time of the joined metals, which have different melting points, limited mutual solubility, and different coefficients of heat capacity, thermal expansion, and thermal conductivity. Because of the high welding speeds and the high depth-to-width ratio of the laser weld seam, the width and volume of the weld pool of most alloys are small compared to arc welding. This contributes to a high cooling rate of the weld seam metal, which in turn can lead to the minimization of the IMC formation when welding dissimilar metals.

The main technological regimes of laser welding are laser power and welding speed. As a rule, the main factors affecting laser power and welding speed are the thickness of the welded workpieces, the wavelength of the laser irradiation, the laser spot diameter, the absorption coefficient of the laser irradiation, and the thermophysical properties of the welded alloy to which the laser beam is directed (melting point, thermal conductivity, heat capacity) when welding dissimilar alloys. For example, the speed of remote laser welding can reach 1000 mm/s, which is tens of times higher than the speed of arc welding. When welding dissimilar alloys, both extremely high speed (800–1000 mm/s) at a power of 5000–7000 W and a laser spot diameter of 50 μm, and very small speed (5–10 mm/s) with heat conduction laser welding of thin sheets in pulse laser mode and power of 100–500 W can be used. The doping level and the absorption coefficient of laser irradiation significantly influence the choice of laser welding parameters. The doping level of the alloy significantly affects the thermal conductivity and heat capacity, for example, when welding the AA5005 alloy, two times more laser power is required than when welding the AA1469 aluminum alloy, taking into account the fact that the sheet thickness, the joint type, and the welding speed are the same. Additionally, the wavelength of laser irradiation has a significant influence on the absorption coefficient of the metal, for example, Al and Cu practically do not absorb laser irradiation of a CO_2_ laser (λ = 10.6 µm). In addition to the above factors, the choice of the main parameters of laser welding (laser power and welding speed) is influenced by the use of other parameters, such as: defocus distance, the inclination angle of the laser beam, laser beam wobbling parameters (if used), type of shielding gas, filler material feed rate, and others.

The purpose of the work is to analyze the methods and techniques of laser welding of dissimilar metals to determine the most optimal and technological approaches when welding such metal pairs as Al/Fe, Al/Ti, Al/Cu.

## 2. Quantitative Analysis of Trends in World Publications

Analysis of publications for 2016–2021 inclusively on the Web of Science abstract database on the topic of laser welding of dissimilar metals in high-rated journals showed that about 200 articles were published. Among them, 43% of publications are dedicated to Al/Fe (26%), Al/Ti (9%), Al/Cu (8%) pairs, and the remaining 57% are dedicated to laser welding of other metal pairs (Figure 1 and Figure 2), including 9% metals + nonmetals. The active use of this metal pair in the automotive and carriage-building industries explains the highest popularity of the Al/Fe pair (26%), as a result of which, in these works describe the technologies for joining sheet blanks with a thickness (up to 2–3 mm). The next most popular pair is Al/Ti (9%); this compound is widely used in aviation and space products to minimize weight. The Al/Cu pair (8%), which is mainly used in the electrical and thermal power industries, is in third place. We should also note the Mg/Ti compounds (5%) that are used in the aircraft and space industries.

## 3. Physical and Thermophysical Properties of Al, Fe, Ti, Cu

Table 2 shows that the physical properties, except for the density, of aluminum, iron, titanium, and copper, such as atomic radius, crystal lattice type, the lattice constant, and electronegativity differ slightly.

Whereas the thermophysical properties of aluminum, iron, titanium, and copper vary significantly, in some cases by tens of times, for example, the thermal conductivity of titanium compared to other metals (Table 3). This suggests that if in the molten state, these metals can interact with each other, then when heated and cooled, the metals will behave significantly differently. For example, a cold crack may form as a result of a significant difference in the coefficient of thermal expansion [18]. When fusion welding, this plays a significant role, especially for types of welding, which are characterized by the formation of a large weld pool [19]. In this case, the minimization of overheating and the volume of the weld pool during laser welding plays a positive role.

## 4. Methods and Techniques of Laser Welding of Dissimilar Metals

Figure 3 shows the configurations of the joint types used for welding dissimilar metals. Compared to the widespread T-joints, butt, fillet, and lap joints used in arc welding, the types of joints for laser welding and welding-brazing have differences and advantages, in particular for joining dissimilar materials. For example, a lap joint or a spike T-joint configuration (Figure 3 “Lap”, “Stake”) has advantages when welding dissimilar metals, in the case when the upper metal melted by a laser beam wets and interacts well with the lower metal and worse the other way round [20]. In the case of edge fillet and single-pass T-joint, using the inclination angle of the laser beam, we can adjust the penetration depth into metal 1 or metal 2, which will affect metallurgical processes. Joint types such as edge or double-flanged edge joints will have advantages when welding-brazing with filler material, or when welding metals, having a low absorption coefficient of laser irradiation, the laser beam will be reflected between the edges, which will increase melting [21]. Additionally, the filler material will be directed into the joint in the form of a groove and the problem of its displacement relative to the laser beam will be solved.

The main techniques and methods used in laser welding of dissimilar metals are:Offset of the laser beam to one of the welded metals;The use of intermediate metals or the application of coatings from a metal that differs from the welded two (Ni, Cu, Au, Ag, and others).

When choosing the offset of the laser beam on one of the welded metals, various factors and properties of the joined metals are used as a guide. The main significantly different properties of metals are the degree of absorption of laser irradiation of a certain wavelength by the metal, the melting point, the mutual solubility of the components at the level of the crystalline structure, the wettability of one component by another, or vice versa, the difference of the coefficient of thermal expansion, heat capacity and thermal conductivity, etc.

For example, when welding well-weldable copper to stainless steel, the laser beam is offset onto the steel, the steel melts, wets and heats the copper through heat transfer in the solid, resulting in metallic bonds. In the case when the laser beam is offset to copper, 99% of laser irradiation of almost all wavelengths in the IR spectrum will be reflected; the thermal conductivity of copper is five-times greater than that of iron [22,23,24] which will lead to the fact that the heat generated by the impact of the laser irradiation on copper will be scattered and will not melt copper and so on.

When welding limited-weldable steel to aluminum, the laser beam is mainly offset to aluminum although its thermal conductivity and the laser beam reflectivity are higher than that of steel, the wettability of steel by molten liquid aluminum is higher than the wettability of aluminum by liquid iron [5]. The melting point of iron is almost three times higher than that of aluminum, which means that the melting of iron can lead to the boiling of aluminum. Additionally, by offsetting the laser beam in the range of 0.1–2 mm, depending on the speed and thickness of the welded workpieces, we can control the thickness of the IMC formation.

The technique of using intermediate metals or the application of coatings that are metallurgically compatible with both metals that are poorly welded to each other became widely used in various types of welding, such as diffusion, explosion, pressure [15]. As a rule, this welding technique is used for lap joints, but in the case of laser welding, it is used for both lap and butts joints. When welding butt joints, the intermediate metal can be melted either directly by the laser beam or as a result of thermal conduction when the laser beam is offset onto one of the welded components. In the case of lap joints, the laser beam does not directly affect the intermediate metal, which is heated up conductively and does not always melt. Most of the joints obtained by the above methods are welded-brazed. In this case, for one metal, the process is characterized as welding, it melts, wetting another metal for which the process is characterized as brazing, the mechanical properties of such joints can reach 70–90% of the properties of a weaker metal [1].

The main advantage of these technological methods is precision control of metal melting through a high degree of controllability of laser welding modes, which makes it possible to control the overheating and thickness of the IMC and, respectively, improve the quality of the joint and its properties.

## 5. Laser Welding of Aluminum Alloys to Steel

The most common laser-welded metal pair is steel and aluminum, as they are the main structural metals. Table 2 and Table 3 present the comparison of the main physical and thermophysical properties of aluminum and iron. From the data in Table 2, we can see that the crystal structure of aluminum and iron is different, the lattice constant differs by almost 1.5 times, the crystal lattice of aluminum is the same only with gamma iron, the atomic radius of aluminum is 143 pm, iron–126 pm. The thermophysical properties of this metal pair vary by almost two times, the thermal expansion and thermal conductivity coefficients, specific heat capacity are higher for aluminum, and the melting point is higher for iron.

Iron has low solubility in solid aluminum and forms eutectic with aluminum. Aluminum dissolves well in alpha iron, forming the following stable phases with a certain region of homogeneity, Fe_3_Al, FeAl_2_, Fe_2_Al_5_, FeAl_3_ [6,14]. Additionally, aluminum is a good deoxidizer and is used in limited quantities in iron-based alloys. Based on the above, fusion welding of aluminum and iron-based alloys is a science-intensive and technological task.

However, using laser welding or welding-brazing, some researchers can obtain quality joints that meet the requirements for the safe operation of vehicles. The mechanical properties of such joints stand at the level of 70% of the used aluminum alloy.

The work [25] presents a technology in which a laser beam was directed to a filler wire, which in the molten state interacts with DX51D steel and AlMgSi_1_ alloy. The authors chose the double-flanged and lap configuration, in this case, the edges being joined by a laser beam do not melt.

In the experiment, various filler wires were used, the main ones were two wires made of an aluminum-based alloy (AlSi_5_, AlSi_12_) and one zinc-based (ZnAl_2_). At tensile tests, the maximum mechanical properties were obtained using specimens with a zinc-based filler material (230 MPa), and specimens with an aluminum-based filler material had (160–180 MPa) (Figure 4). In this case, the destruction of specimens with (AlSi_12_, ZnAl_2_) filler materials was observed along the HAZ of aluminum.

When welding using the butt configuration of the aluminum alloy 6061-T6 and steel DP590 joint, the authors of [26] investigated the effect of the shape of the edge preparation of steel and aluminum (Figure 5a–c), using a filler material based on AlSi_12_. The main parameters of laser welding-brazing are as follows: laser power 2200 W, sufficiently low welding speed 0.5 m/min, defocus distance +40 mm, the distance of laser spot offset to Al 0.4 mm. In addition to mechanical tests of joints, the authors compared the obtained results with mathematical modeling of the thermal field distribution depending on the shape of the edge preparation of steel and aluminum. The temperature field distribution showed that in the case of the half V-shape groove on the side of the steel, there was a tendency towards a more uniform thermal field distribution. The tensile strength of the test specimens was 108–145 MPa, the elongation was less than 1 mm, the specimens with the groove shape shown in Figure 5c had the highest values, and they also had the minimum IMC thickness (8.8 μm). Whereas the samples with the groove shape (Figure 5a) had the smallest value of the tensile strength, they also had the greatest IMC thickness. It suggests that the melting and wetting of the lower edge of the steel with a liquid filler material has a positive effect on the mechanical properties of the joint of the sample with the half V-shape groove. The authors of [27] came to the same conclusions when studying butt-welded joints of titanium and aluminum.

The works [2,28,29] describe the studies of lap welding of aluminum and steel sheets. The authors of [2] investigate the influence of heat input on the microstructure of welded joints when welding an aluminum-based and iron-based alloy under the impact of a beam from the side of aluminum or the side of the steel. Figure 6 shows the microstructure of the joints obtained by the impact from the side of the steel Figure 6a–c and the side of the aluminum Figure 6d–f. As you can see in the presented photos (Figure 6a,f), the penetration of steel is significantly larger under the impact of a laser beam with a heat input of 45.00 kJ/m. As a result of the carried out studies, the authors conclude that under the impact of a laser beam from the side of the aluminum, liquid aluminum actively interacts with steel. It leads to the formation of IMCs of high thickness, and in some cases, to the formation of cracks. Whereas under the impact of a laser beam from the side of the steel, the amount of molten aluminum is controlled better, by which means the thickness of the formed IMC can be controlled.

The authors of [28] describe the modeling of the propagation of temperature fields under the lap laser welding of steel and aluminum by a defocused beam with a diameter of 13 mm. The modeling results of the welding process show that the established boundary conditions and the applied model correspond to the results of a real experiment–depth and width of molten steel and aluminum (Figure 7). Created by a defocused laser beam, the maximum width of the interface between steel and aluminum and the minimum IMC thickness provide maximum mechanical properties for shear testing.

The work [29] describes the studies of lap welding of steel and aluminum with a split laser beam. The laser beam impacts from the side of the steel, the beam is split along or across the welding direction, and the distance between the beams and the power ratio of the laser beams are also varied. The maximum mechanical properties in shear tests (109.2 N/mm) were obtained with the transverse arrangement of the laser beams relative to the welding direction and the power ratio of the beams 3/2 (Figure 8). The authors also established that at the weld seam boundary, both from the steel side and the aluminum side, the same IMC phases–η-Fe_2_Al_5_, θ-Fe_4_Al_13_ are observed. However, the shapes of the phases differ on the side of the steel grain boundaries; they are very fine, on the side of the Al/weld interface–needle-like grains, and closer to the base metal of aluminum–fine grains.

Hybrid laser-arc welding, in which the main variable parameter is the distance between the laser beam and the electric arc and the effect of its thermal cycle, is also used for butt-joining steel to aluminum.

The authors of [30] carried out the comparison of the two techniques–the offset of the laser beam on steel and hybrid laser-arc welding (the arc and the beam are directed into the butt-joint). A less defective joint was obtained using a laser offset welding to the steel, which is accompanied by higher cooling rates and, as a result, the formation of IMCs of the smaller thickness (6 μm). Hybrid laser-arc welded joints had more defects as a result of the formation of a large weld pool and a significant difference in thermal- and fluid-dynamic properties of the two metals.

The work in [31] describes the technology of laser-arc welding-brazing with an intermediate material based on aluminum (Al80Zn8Mg7Mn2Si2) pressed from a powder. The laser beam and arc are directed to the intermediate material, and the main variable parameters are laser power, welding speed, distance from the center of the arc torch and laser beam, and arc current. As a result of the carried out research, the authors come to the following conclusions. The use of a laser beam and an arc as two heat sources increases the spreading of the intermediate material. However, with an increase in laser power, arc current, and welding speed, the IMC thickness increases, and an increase in the distance between two heat sources leads to a decrease of the IMC thickness. The formed welded-brazed joint has a tensile strength of 163 MPa and an IMC thickness of 8.7 µm (Figure 9).

The authors [32] proposed the technology of T-joints production based on the thermomechanical interaction. The peculiarity of the technology lies in the fact that an aluminum sheet is inserted into a groove on a steel sheet then a defocused laser beam heats the reverse side of the steel sheet along the groove trajectory (Figure 10a). During the experiment, the power and exposure time of the laser beam are varied. The modes in which the steel sheet is heated to temperatures sufficient to melt the aluminum sheet edge are optimal. As a result, by wetting the steel with molten aluminum, metal bonds are formed between the end face of the aluminum sheet and the metal of the groove cavity of the steel sheet. In addition to metallurgical contact between steel and aluminum, a mechanical inter-locking effect of the aluminum sheet with the metal of the groove cavity occurs (Figure 10a,b), which increases the mechanical characteristics of the joint. In this case, the thickness of the resulting IMC is about 5 μm.

The work [33] investigates the influence of laser beam offset and laser beam circular oscillations on the behavior of molten aluminum, metallurgical processes, and, as a result, on the microstructure and mechanical properties of the welded joint. The experimenters use a laser beam (Laser power (P) of 2.4 kW) and an electric arc (arc current (I) of 58 A, and arc voltage (U) of 15.5 V), welding speed 2 m/min, oscillating frequency 150 Hz, and oscillating amplitude 1 mm as energy source. Only the laser beam offset (∆D = 0.2; 0.4; 0.6; 0.8; 1.0 mm) was varied from the side of the aluminum (Figure 11).

Figure 12 shows a schematic representation of the resulting welded-brazed joint. As mentioned above, when the laser beam and the electric arc are offset onto aluminum, it melts and wets the steel. The authors pay attention to the study of the top, middle, and bottom zones of the joint microstructure (Figure 12), depending on the laser beam offset (∆D).

Usually, when melting aluminum-based alloys of small thicknesses (up to 2–3 mm), a cone-shaped weld pool is formed by a laser beam. By using oscillations of the laser beam along a circular trajectory and with a particular frequency, a cylindrical weld pool will be formed. A weld pool of this shape will provide a more uniform interaction in the depth of molten aluminum with the steel edge. As a result, a uniform transitional IMC will be formed from the weld pool to steel. And using the laser beam offset, the IMC can be minimized. The authors managed to select the welding modes in such a way that at laser beam offset ∆D = 0.8 mm, the thickness of one Al_5_Fe_2_ IMC was ~1.3 μm. The tensile strength of such joints was about 160 MPa. Whereas at the laser beam offset ∆D = 0.4 mm, the thickness of the top IMC differed from the thickness of the bottom IMC by a factor of 29 (0.9 μm and 25.8 μm, respectively), and the bottom IMC consisted of several layers and phases of Al_5_Fe_2_, Al_13_Fe_4_, the tensile strength of such joints was 77 MPa. Figure 13 presents the microstructure of the top, middle and bottom FZ/304SS interface depending on the laser beam offset ∆D = 0.6; 0.8; 1.0. Two-phase IMC (Al_5_Fe_2_, Al_13_Fe_4_) is observed only in the middle and bottom parts of the sample at the laser beam offset ∆D = 0.6.

Other papers on the topic of laser welding of steel and aluminum describe the choice of filler material [34,35,36,37], shielding gas [38], the application of coatings on the metals to be joined [35,36,37,38,39,40,41], the use of double-beam laser welding [42,43,44], or the use of remote high-speed welding [45,46] (Table 4).

From the presented data, we can draw the following conclusions about the laser welding of steels and aluminum alloys. The main advantage of laser welding is the precision of the action and, as a consequence, the minimization of the interaction time of aluminum and iron in the liquid state. This circumstance leads to minimization of the IMC formation. The basic techniques and methods used in butt and lap welding of steel and aluminum are different.

When welding a lap joint or a spike T-joint configuration, the laser beam impact from the side of the steel, which will heat or melt and, through thermal conduction, will heat the aluminum to moderate temperatures (below the boiling point), is preferred. As a result, a minimum zone of interaction between aluminum and steel will be formed, which can be controlled by the parameters of welding modes (welding speed, laser power).

For butt-welding of steel and aluminum, the most preferable is the laser beam impact from the side of the aluminum [47,48]. Although aluminum has a higher thermal conductivity and a lower absorption coefficient of laser irradiation, aluminum has a lower melting point and better wettability of solid steel in the liquid state. The tensile strength of joints obtained using such techniques reaches 150–160 MPa (75–85% of the base aluminum metal); such properties are acceptable for some structures [49,50,51,52,53,54,55,56]. Additionally, the wettability of the upper and lower edges [18,26,27] and the IMC uniformity over the entire depth of the weld seam [33] play an important role for the mechanical properties of the joint. That can be controlled by the precision of the beam offset to aluminum, the choice of filler material, and other technological parameters of laser welding without the use of complex technologies of controlled heating and cooling or the use of additional energy sources.

When welding-brazing dissimilar metals with filler material for such types of joints as double-flanged edge or flare, it is rational to direct the laser beam to the filler material, which will melt and wet the edges of the joined materials.

## 6. Laser Welding of Aluminum Alloys to Titanium Alloys

As mentioned earlier in the Introduction, the second most common pair of metals welded by a laser beam is titanium and aluminum. Obtaining a durable fixed joint of this pair of metals has been an urgent task since the beginning of spaceships construction and modern aviation. At that time, attempts of heat-press welding of titanium and aluminum adapters were made; at present, other types of mechanical and thermomechanical welding classes are actively used [57,58,59]. The resulting welded joints were used to manufacture various aircraft systems that were passing the tests successfully.

Table 2 shows that some physical properties of aluminum and titanium differ significantly, the atomic radius differs by several percent (Al–143 pm, Ti–147 pm), crystal lattices differ (Al–FCC, Ti–HCP, BCC). Whereas the thermophysical properties differ several times, and some-dozens of times, for example, the thermal conductivity coefficient (Al–238 W/(m × K), Ti–22 W/(m × K)).

In the solubility system of titanium-aluminum, three peritectic reactions occur: At a temperature of 1460 °C with the formation of a *γ*-phase, TiAl;At a temperature of 1340 °C with the formation of TiAl_3_,At a temperature of 650 °C with the formation of a solid titanium solution based on aluminum from a melt containing 0.15 wt % of Ti [20]. When melting these metals, aluminum in titanium forms bounded regions of 𝛼 and 𝛽 solutions, as well as intermetallic phases [17].

When welding titanium and aluminum, usage of a laser beam has advantages compared to other sources of energy and melting. Because of the high degree of parameters control of laser welding modes, such as power and welding speed, it is possible to precisely control the melting volume of metals and, accordingly, their metallurgical interaction. This can be used to control the volume of IMC formation and composition. Similar to welding of steel and aluminum, when welding titanium and aluminum, butt and lap joints are most common.

It should be noted that in the analysis of articles on laser welding of steel and aluminum, there were no papers on post-welding heat treatment of these joints [60], while some researchers used heat treatment for welded joints of titanium and aluminum.

The paper [61] investigates the effect of the laser beam offset when welding titanium alloy VT-20 with aluminum alloy AA1461. The joint type is butt joint; when welding, the laser beam was offset by 0, 0.5, and 1 mm onto the titanium alloy, and the workpiece thickness was 2 mm. After obtaining the joint, the researchers carried out heat treatment of the welded joints at temperatures of 490, 540, and 590 °C, for 4 and 6 h. The results of mechanical tests showed that specimens without heat treatment with a laser beam offset by 1 mm had the maximum values of the tensile strength. The use of heat treatment led to a decrease of the mechanical properties, and at a temperature of 590 °C, the tensile strength of the obtained samples with the laser beam offset by 0 and 0.5 mm decreased to 0 MPa. And the samples obtained with the laser beam offset by 1 mm showed the maximum mechanical properties after heat treatment at a temperature of 490 °C (168 MPa).

Such a critical decrease of mechanical properties after high-temperature heat treatment (540–590 °C) can be explained by a significantly different coefficient of thermal expansion (Table 3, Al–23.5, Ti–8.9 10^6^ K). That is, the expansion of one of the metals during heating led to the destruction of the brittle transition IMC layer. Unfortunately, the authors do not provide photographs of the microstructure after fracture to observe the region in which the fracture occurred.

The authors of [62] investigate the effect of low-temperature heat treatment on the mechanical properties of welded joints and the IMC thickness of titanium (Ti-6Al-4V) and aluminum (AA5754) laser-welded joints. The joint type is butt joint; the laser beam was offset to titanium by 1 mm, two types of welded joints were obtained with linear energy 36 and 72 J/mm (Figure 14). Heat treatment of the obtained joints was carried out for 336 and 138 h at a temperature of 350 and 450 °C, respectively.

The paper presents the research results of the chemical and phase composition using X-ray diffraction, the nanohardness of various IMCs, the effect of heat treatment on the change in the grain size of the AA5754 base metal, and fractography of the obtained samples. The IMCs formed in the welded joints had a thickness of 30–50 μm, depending on the heat input during welding; the smallest thickness was typical for the samples obtained with the lowest heat input. In the Conclusions section, the authors conclude that the heat treatment at a temperature of 350 °C does not affect the tensile strength (90–100 MPa), which is comparable with the maximum one for specimens without heat treatment (100–110 MPa), while the elongation slightly increased. Additionally, heat treatment at a temperature of 350 °C does not affect the IMC thickness and structure. Whereas after heat treatment at a temperature of 450 °C, the IMC thickness increased, and the tensile strength decreased to 55–75 MPa.

The paper [27] describes the study of the effect of laser beam offset on aluminum alloy 5A06 when butt welding the 1.5 mm thick sheets of titanium alloy Ti6Al4V and aluminum alloy 5A06. Variable parameters during the experiment are Laser power (1130–1280 W), Welding speed (8–11 mm/s), Laser offset (300–700 µm). Welded joints with a maximum tensile strength (183 MPa), which is 83% of that of aluminum alloy, with the IMC thickness of about 3 µm were obtained when using welding modes with the following parameters: Laser power (1130 W), Welding speed (11 mm/s), Laser offset (500 µm). The authors paid special attention to the study of the spreading of the molten aluminum over the top and bottom parts of the titanium sheet. Additionally, the fractures of the obtained joints were investigated after tensile tests, in particular the chemical composition of the top, middle and bottom parts. In samples with good aluminum wettability of the top and bottom parts (Figure 15a,b), the chemical and phase composition of the middle part was significantly different from that of the top and bottom parts. When using some modes, samples with a lack of fusion and bottom spreading (Figure 15c) were obtained, the mechanical properties of which were lower than those of samples with good wettability of the top and bottom parts.

The top and bottom parts of the weld seam have a phase composition that is substantially different from the middle part because they are formed as a result of a longer interaction of molten aluminum and titanium. The IMC thickness in depth also differs significantly (Figure 16a,b).

In the Conclusions section, the authors conclude the following. By varying the mentioned main parameters of welding modes, it is possible to obtain a high-quality joint of titanium and aluminum with high mechanical properties. With an increase of the laser beam offset to aluminum and a decrease of heat input (an increase of the speed and a decrease of the laser power), it is possible to reduce the IMC thickness. However, this can lead to a decrease of the wettability of the bottom part. Good and uniform wettability of the top and bottom parts of the titanium sheet has a positive effect on the mechanical properties of the obtained joint.

The authors of [63] investigate the influence of the parameters of laser butt welding modes of titanium alloy Ti6Al4V and aluminum alloy AA6061 with a thickness of 4.5 mm and 5 mm, respectively. The main variable parameters of welding regimes are Laser power (1.5–3.5 kW), Laser beam oscillation frequency (25–30 Hz), Laser offset (1.1–1.2 mm), the laser beam was offset to an aluminum alloy, the beam oscillation trajectory is sinusoidal, the welding speed is constant 1 m/min. These parameters, varied in a rather narrow “parameter window”, significantly influence the microstructure and mechanical properties of the obtained joints. When using the offset of 1.1 mm with the laser beam oscillation frequencies of 25 and 30 Hz, the tensile strength of the joints was 128–139 MPa; with an offset of 1.2 mm and the same oscillation parameters, the tensile strength increases to 164–173 Mpa, with the IMC thickness of ~2 μm. Whereas, at the offset of 1.1 mm, the IMC thickness varied within 1.6–5.6 μm, depending on the oscillation frequency and the weld seam section in-depth (top, middle, bottom).

Nikulina A.A. et al. in [64] use significantly higher laser welding speeds (70–100 mm/s) in comparison with the above works (11–17 mm/s) [45,46,47,48]. The effect of the welding speed and the level of heat input when butt welding the titanium alloy VT6S (3 mm thick) and aluminum alloy AA1424 (4 mm thick) was investigated. The laser beam was offset to an aluminum alloy by 200 μm. Microhardness, phase and chemical composition of the IMC were intimately examined. The authors conclude that with an increase of speed (100 mm/s) and a decrease of heat input, a thinner IMC is observed (less than 1 μm).

The work [65] describes the influence of the parameters of pulsed laser welding modes on the microstructure and mechanical properties of joints made of titanium alloy Ti6Al4V and aluminum alloy AA6060. The joint type was lap joint, the beam impacted on the titanium alloy, the maximum laser power was 300 W, welding was performed with overlapping of laser points. The authors carried out a profound study of interfacial crack initiation regions, which are an important indicator of mechanical properties. It was revealed that a surface with a high level of dislocation density between different IMCs (TiAl and TiAl_3_) is a probable place for the initiation and propagation of cracks. The results of the conducted studies showed that the IMC of constant composition will be more stable in terms of mechanical properties.

The works [17,20] investigate the influence of the technique and parameters of the laser welding modes on the metallurgical processes and the microstructure of the welded seams of the lap joints. In both works, the authors investigate the impact of a laser beam, both from the side of titanium and the side of aluminum, which significantly influences the mechanism of the joint formation, and, as a result, the microstructure and properties of the joints. The work [17] investigates in detail the IMC formation and type depending on the welding speed (5–50 m/min) and the impact from the side of aluminum or the side of titanium. The authors conclude that increasing the welding speed to 50 m/min reduces the probability of formation and the number of brittle IMCs Al_3_Ti, Al_2_Ti, which is caused by high cooling rates that suppress the formation of a large volume of IMCs.

The work in [20] also investigates the effect of welding modes and the impact from the side of the titanium (CP Ti) and the side of the aluminum (AA2024) at 2.5 kW laser power and 120–150 mm/s welding speed. Under the impact of the laser beam from the aluminum side, poor wettability of the titanium alloy with molten aluminum was observed. Under the impact from the side of titanium, a stable formation of a welded joint was observed. The composition and structure of the joints depended on the level of heat input, in particular, the chemical and phase composition of the transition layers differed and, as a result, the mechanical properties. When studying the microstructure after tensile shear strength testing, the fracture was observed not only over the IMC, as it usually happens, but also over the heat-affected zone (HAZ) of aluminum (Figure 17a,b). More precisely, the HAZ of aluminum consisted of two parts columnar and fine-grained, and the cracks were observed along the line between these two HAZ in all studied samples.

From the above, it follows that in [60,61,62,63], experimenters used low laser welding speeds of 11–17 mm/s, and in works [17,20,64,65,66], the used welding speeds were tens of times higher (100–833 mm/s). At the same time, as a result of a higher laser beam offset on the aluminum alloy, the IMC is comparable in both cases (2–6 μm). As a result of the analysis of the above articles on welding of alloys based on titanium and aluminum, the following conclusions can be drawn. The same as when welding steels with aluminum alloys, the IMC composition and thickness are the weakest part of the welded joint at mechanical tests. When butt welding, the basic technique is to offset the laser beam onto one of the alloys. By offsetting the laser beam on aluminum, it is possible to obtain the minimum IMC and maximum mechanical properties. The mechanical properties of welded joints, obtained by offsetting the laser beam on a titanium alloy, are 40–50% lower than when offsetting the laser beam on an aluminum alloy. This significant difference is explained by the fact that when offsetting the laser beam on an aluminum alloy, the TiAl and TiAl_3_ IMCs are formed, and when offsetting the laser beam on a titanium alloy, more brittle Al_3_Ti, Al_2_Ti IMCs are formed. High welding speeds (200–300 mm/s) and a decrease of heat input contribute to the minimization of the IMC thickness.

When lap welding, it is reasonable to impact from the side of titanium since molten titanium better interacts with solid aluminum than molten aluminum with solid titanium.

## 7. Laser Welding of Aluminum Alloys to Copper

Copper and aluminum form solid substitution solutions of limited solubility; the limiting solubility of copper in aluminum is 2.2 at %. This is justified by the fact that the lattice parameters and the size of copper and aluminum atoms are similar, the type of the crystal lattice is the same, and the lattice constants are comparable (Table 2). Additionally, copper and aluminum form a series of intermetallic phases (Al_2_Cu–θ, AlCu–η_2_, Al_3_Cu_4_–ζ_1_, Al_4_Cu_9_–δ, Al_4_Cu_9_–γ_1_) [57].

Copper and aluminum are widespread conductors of electric current; therefore, practically pure alloys with minimal electrical resistance are mainly used in welding. Both copper and aluminum have a very low absorption coefficient of laser irradiation in the IR spectrum, high values of thermal conductivity, which complicates the process of laser welding. However, due to the fact that the laser beam is multifunctional, as a tool for metalworking, it is sometimes used for welding copper and aluminum. Laser micro-welding of electrical contacts made of copper and aluminum has been used for a long time and till the present day. As a rule, such joints require metallurgical contact, and there are no requirements for high mechanical properties.

The following studies of laser-welded joints describe the formed types of IMCs and their effect on mechanical properties and electrical resistance. Typically, the IMC electrical resistance is 5 to 8 times higher than that of copper or aluminum.

In [67] using EDX and micro-XRD, the authors investigate in detail transitional IMC of samples of Al-Cu welded joints obtained by laser welding. Laser beam welding is performed with an overlap under the impact from the side of the aluminum side. Compositions and type of IMCs, such as Al_2_Cu–θ, AlCu–η_2_, Al_3_Cu_4_–ζ_1_, Al_4_Cu_9_–γ_1_, were investigated in detail, depending on the section of the transition of the weld seam from aluminum to copper to determine the most brittle layer. After determining the chemical composition of the fracture surface (61.6–62.7% copper, the rest is aluminum, which corresponds to the Al_4_Cu_9_–δ phase), the authors concluded that cracks generally originated in the AlCu and Al_4_Cu_9_ phases.

The work [68] describes the study of the influence of oscillations and laser beam offset on aluminum when butt welding the copper (99.9% Cu) and aluminum (99.5% Al). The laser beam offset was performed in the range ∆x = 0–400 µm, circular 2D beam oscillation was used, welding was carried out in a vacuum at reduced atmospheric pressure. The authors also measured the electrical resistance of the obtained welded joints and compared them with those measured for copper (99.9% Cu) and aluminum (99.5% Al). As a result of the carried out studies, the authors showed that under the laser beam offset of 100 µm on aluminum, the IMC was 3–4 times larger than under the laser beam offset of 200 µm (Figure 18). So the melting ratio of the joined materials is an important factor that can be successfully controlled by the laser beam offset for obtaining a high-quality joint. The minimum interaction layer between copper and aluminum was 80 μm, the IMC thickness was 8–13 μm, which were obtained by offsetting the laser beam on aluminum by 300 μm.

The authors of [69] investigate the influence of the amplitude and frequency of a laser beam sinusoidal oscillations perpendicular to the welding direction. Aluminum (Al 99.5) and pure copper (Cu–OF) with a thickness of 1 mm were lap weld under the impact of the laser beam on the aluminum. The authors investigated the influence of the parameters of welding modes on the mechanical properties and electrical resistance of the obtained joints. The parameters of the welding regimes were as follows: laser power P = 3.25 kW, welding speed–6 m/min, the oscillation frequency varied from 200 to 1000 Hz, and the oscillation amplitude varied from 0.25 to 1.00 mm. Oscillations of the laser beam transverse to the weld seam led to an increase of the interaction region of aluminum and copper to 1.0–1.2 mm, while without oscillations, the width of the interaction region was 0.2–0.3 mm (Figure 19a,b). The paper considers in detail the influence of the amplitude and frequency of oscillations on the weld seam geometry, copper content in the weld seam, electrical resistance, and mechanical properties. After analyzing a larger volume of experimental data, the authors made the following conclusions. When lap welding the 1 mm thick workpieces, to ensure the minimum electrical resistance, the width of the interaction region should be at least 1 mm (Figure 19a). When the width to depth ratio of the weld seam is >4 (Figure 19b) and the depth of penetration into the copper alloy is >0.2 mm, the minimum electrical resistance and maximum mechanical properties can be obtained. Based on the correlation between the values of electrical resistance and mechanical properties, the authors proposed to use the measurement of electrical resistance as a nondestructive testing method of such joints.

When butt-welding the dissimilar metals, the accuracy of the laser beam offset on one of the welded workpieces is a very important technological parameter. The interaction region of the welded workpieces and the IMC thickness significantly depend on the offset accuracy in the range of 50–100 μm. The authors of [70] proposed a control method for the micro-offsets of a laser beam based on observing the change in the wavelength of the emission spectrum of the fiber laser induced plasma, depending on the laser beam offset on one of the welded workpieces. As a result of the conducted studies, the authors established that the specific peaks of the wavelengths for aluminum (394.4 and 396.1 nm) and copper (578 nm) could be used as a new control method of the laser beam offset, which is cheaper than using a spectrometer. 

When analyzing publications, it was found that after laser welding, most researchers conduct mechanical tests (mainly tensile or shear strength tests) [71,72,73,74,75,76,77] while vital Al/Cu joints properties are thermal conductivity and electrical conductivity [78]. That is, the main area of application is the branch of electrical contact or heat transfer products and devices. Of course, some joints may be subjected to mechanical stresses, in particular cyclic ones with a small amplitude, the resistance against which can be ensured by mechanical fixation of parts welded from Al/Cu. In this case, it is necessary to pay great attention to the study of operational properties, in particular, electrical conductivity or degradation and changes in the IMC during electrical usage.

Especially it is necessary to highlight Reisgen et al. [79] investigating the operational properties of the Al/Cu welded joint when a current of 200 amperes passes through it, which is a more in-depth continuation of the work describing the production of an Al/Cu welded joint [68]. Electrical usage was carried out for two weeks, after which the study did not find an increase in the IMC.

Another group of researchers led by Schmalen studied the robustness of the laser braze-welding process [80] and the measurement of the resistance of the Al/Cu overlap joint by the 4-wire method [81]. In this work [81], a circular laser beam wobbling was applied to increase the contact area of dissimilar metals, and the effect of laser power on electrical resistance was estimated, which is similar to the approach described in [69]. The optimal laser power is in the range of 900–1400 W.

A significant advantage of laser welding, which allows reducing the thickness of the IMC, is the use of high-speed laser welding up to 95–155 mm/s [75], 250 mm/s [71], 80–800 mm/s [76], which makes it possible to obtain a minimum IMC with a thickness of 1–4 μm with heterogeneous dual-phases (copper and γ-Al_4_Cu_9_ phases), which is also formed at solid-state welding [82]. The recently published work [71] on laser pressure welding is of particular interest, a process that combines laser welding with rolling welding, with a laser beam directed between two converging sheets of aluminum and copper. The authors of this work were the first to use this technique for welding the pure aluminum, then for pure copper, which are known to reflect laser irradiation strongly, and then for their combination. It should be noted that the use of longitudinal (relative to the welded plane of the sheets) laser beam wobbling, in this case, would significantly increase the area of the welded joint the tensile shear strength of the joint reached 111.94 MPa.

After tensile tests, failure areas of the Al/Cu welded joint were mainly observed in the copper and γ-Al_4_Cu_9_ [67,71,83] phase and the θ-CuAl_2_ phase [75,84]. In comparison with articles on laser welding of Al/Fe, Al/Ti, in articles on Al/Cu welding, little attention is paid to the hypothesis of the welded joint formation, in particular, the formation of various IMC phases and modeling of welding processes and their comparison with experimental data, which is an effective method of scientific research.

Additionally, several other works, in which micro-welding of electrical contacts [85,86,87,88], the influence of the choice of filler material and welding modes on the microstructure and mechanical properties of welded joints [72,89,90,91,92], studies of the optimal offset of the heat source based on the diagram phase state [93] are considered, are dedicated to laser welding of aluminum and copper.

Making conclusions on the conducted analysis of publications on laser welding of aluminum and copper, we can conclude the following. When butt welding, the laser beam is offset on aluminum, which allows obtaining the minimal IMC with satisfactory electrical properties. Additionally, the selection of filler material can influence the wettability properties during welding, and the resulting electrical and mechanical properties.

The main areas of research in the field of welding Al/Cu metals on the topic of which there are still not enough publications are surface modification before welding in order to increase wettability, the use of remote laser welding, which allows using ultra-high welding speeds, a more thorough study of the electrical conductivity of the obtained welded joints, as well as assessment of the economic efficiency of using the resulting welded joints. A serious challenge is the use of shorter wavelength lasers, the irradiation of which (1–2 kW) is difficult to focus at one spot to obtain maximum power density. In this case, it will be relevant to use laser beam wobbling systems, which allow heating the metal quickly and distributing the used small irradiation power evenly.

When lap welding, the laser beam impact from the side of the aluminum is reasonable. It is possible to obtain welded joints of aluminum and copper with the tensile shear strength of 95–128 MPa using this technique. In this case, the width of the weld pool can be increased by defocusing or transverse wobbling (oscillation) of the laser beam to increase the electrical conductivity over the contact area of aluminum and copper.

## 8. Discussion and Conclusions

The use of welded joints from dissimilar metals in all industries is a promising direction and will help to reduce the weight of structures and vehicles, increase their carrying capacity, and, accordingly, reduce harmful emissions into the environment. Laser welding is a promising method for joining dissimilar metals due to its precision of action when aiming a laser beam (with an accuracy of 0.1 mm), the ability to accurately control heat input, especially using a laser spot of small diameter (50–70 μm).

The main weak point of laser welded joints of pairs (Al/Fe, Al/Ti, Al/Cu) is IMC, which has high hardness and brittleness, electrical resistance. The main directions of scientific research in the field of laser welding of dissimilar metals are the minimization of the IMC, the choice of filler material and the study of the IMC composition, the nature, and conditions of the IMC formation.

The main directions of scientific research in the field of laser welding of dissimilar metals are the minimization of the IMC, the choice of filler material and the study of the IMC composition, the nature, and conditions of the IMC formation. Since the speeds of laser welding are sufficiently high (up to 200 mm/s), the use of wire filler material for butt welding of small thicknesses (up to 2 mm) will be difficult because the wire will not always have time to melt. In the case of an overlap joint, it is impossible to use the wire as a filler material, therefore the use of thin plates or electroplating of thin layers (10–50 μm) of metals (Cu, Ni, Ag) on one of the welded dissimilar metals will be promising [94]. The main method for studying the composition of the IMC, which used by almost all authors of articles, is SEM using the EDS and EBSD technique with the help of which it is possible to identify the possible phase composition of the IMC.

Obtaining permanent joints from dissimilar metals is an urgent challenge, not only in the field of laser welding of dissimilar metals but also in such promising and science-intensive manufacturing areas as additive technologies (Direct Metal Deposition, (DMD), Selective Laser Melting (SLM)) [95,96,97,98] and technology of recovery cladding [99,100]. They also have problems with cracking, poor wettability, and flowability of the molten metal. As a rule, the substrate or the parts to be recovered differ in thermophysical and metallurgical properties from the metal being deposited. Therefore, the techniques and methods used in laser welding of dissimilar metals can be used in additive and cladding technologies.

Analysis of 2021 publications shows that promising areas are modeling [101,102], the use of wobbling [103,104,105], bilateral laser welding [106], the application effect of preliminary preparation of the titanium surface on its wettability with molten aluminum [107]. When studying the metal of the transition layer of welded joints from dissimilar alloys, the EBSD and XRD methods are used to reveal the phase composition and high-speed video recording of laser welding of dissimilar metals [108]. The authors of [109] proposed to use a titanium-aluminum joint in an aircraft wing structure, where titanium is used as a skin and is laser-welded to a structural frame made of aluminum, which will reduce the weight of the structure.

In [102], a simulation of the welding process of dissimilar Al/Ti alloys was carried out when studying the joint microstructure, the authors obtained results very similar to my results presented in [20]. In [20,102], the aluminum alloy AA2024 is used, and authors of both works identified two different heat-affected zones from the aluminum side; the microhardness measurements also showed similar values. It says the repeatability of experimental results by absolutely various researchers.

A quantitative analysis of 2021 papers made according to the WoS system showed that there were almost as many articles on the subject of Al/Ti laser welding as there were on Al/Fe laser welding (Table 5). Whereas in previous years, this difference was almost three times greater in favor of Al/Fe. This indicates a sufficient study of the Al/Fe laser welding topic and insufficient knowledge of the Al/Ti laser welding topic. Moreover, the techniques and methods for increasing the Al/Ti weldability are similar to that of Al/Fe and borrowed from the studies of Al/Fe welding.

An interesting fact was noted in the analysis of 2021 publications. Three groups of scientists continue more in-depth studies of the results already obtained by them or other scientists. Katayama et al. [108] continued research [17] on high-speed laser welding of titanium and aluminum (50 m/min) using a high-speed camera, and a more detailed study using in-situ spectroscopic analysis of plume induced during Al (upper)–Ti (lower) and Ti (upper)–Al (lower) dissimilar welding. Casalino G. et al. [110] continue their studies [62] in this case, in addition to the offset of the laser beam, “-” defocusing, heat treatment, and ultrasonic peening are used. This allows obtaining the mechanical properties of the joint at the level of 173 MPa and the elongation of 4.5%. Gao M. et al. in [105] continue similar studies of the authors of [33] on Al/Fe also aimed at studying the offset of the laser beam on aluminum (∆D), the effect of the oscillation frequency on the interface formation (Al_5_Fe_2_), hydrodynamic processes in the molten pool, in particular on the uniform distribution of Fe melt by stirring effects on the molten pool.

In other works, the influence of various filler materials [111,112] and intermediate interlayer AlSi_12_ [113] or Nb [114] was considered. It should be noted that in almost all works of 2021, the authors put forward hypotheses of the formation of a joint at the atomic level. It is based on a deep study of the microstructure of the resulting IMC layer.

As can be seen from Table 5, the minimum laser welding speeds are used for welding the Al/Fe pair, the maximum ones—for the Al/Ti pair. The maximum laser power is used when welding the Al/Fe pair, which is justified by the thermophysical properties of the welded alloys and the joint configuration, as mentioned in the introduction.

The conducted analysis of publications on the topics of laser welding of dissimilar materials (Al/Fe, Al/Ti, Al/Cu)–methods and techniques (microstructure and properties) allows us to draw the following conclusions.

Laser welding of aluminum-based alloys and steels is a highly efficient and promising technology. The use of the precision of laser welding allows accurately controlling the interaction of the welded metals and, as a result, influencing the minimization of the IMC formation. When welding the lap joint or spike T-joint configuration, the impact of the laser beam on the steel, which will be heated or melted, and through thermal conduction will heat the aluminum, is preferable. As a result, a minimum zone of interaction between aluminum and steel will be formed, which can be controlled by the parameters of welding modes (welding speed, laser power).The impact of the laser beam on the aluminum is the most preferable when butt welding the steel and aluminum. The tensile strength of joints obtained using such techniques reaches 150–160 MPa (75–85% of the tensile strength of the base material–aluminum); such properties are acceptable for some structures.When welding-brazing dissimilar metals with filler material of such joint types as edge, double-flanged edge, or flare joints, it is rational to direct the laser beam on the filler material, which will be melted and wet the edges of the joined materials.When welding titanium alloys with aluminum alloys, the IMC composition and thickness are the weakest part of the welded joint during mechanical tests. When the laser beam is offset on aluminum during butt welding, it is possible to obtain the minimum IMC and maximum mechanical properties. The mechanical properties of welded joints obtained by the laser beam offset on titanium alloy are 40–50% lower than when the laser beam offset on an aluminum alloy. This significant difference is explained by the fact that when offsetting the laser beam on an aluminum alloy, the TiAl and TiAl_3_ IMCs are formed, and when offsetting the laser beam on a titanium alloy, more brittle Al_3_Ti, Al_2_Ti IMCs are formed. High welding speeds (200–300 mm/s) and a decrease of heat input contribute to the minimization of the IMC thickness. When lap welding, it is reasonable to impact from the side of titanium since molten titanium better interacts with solid aluminum than molten aluminum with solid titanium.When butt welding aluminum and copper, the laser beam is offset on aluminum, which allows obtaining the minimal IMC with satisfactory electrical properties. Additionally, the selection of filler material can influence the wettability properties during welding, and the resulting electrical and mechanical properties.When lap welding aluminum and copper, the laser beam impact from the side of aluminum is reasonable. It is possible to obtain welded joints of aluminum and copper with the tensile shear strength of 95–128 MPa using this technique. In this case, the width of the weld pool can be increased by defocusing or transverse wobbling (oscillation) of the laser beam to increase the electrical conductivity over the contact area of aluminum and copper.It should be noted that when welding the Al/Cu pair, the main requirement to the joint is to ensure a minimum electrical resistance. Whereas when welding Al/Fe, Al/Ti, maximum mechanical properties are required. It should be noted that the processes modeling of the laser welding of dissimilar materials is widespread, and the obtained data are verifiable [115,119]. However, there are few articles on the modeling of laser welding of Al/Fe, Al/Ti, Al/Cu in open literature sources.

## Figures and Tables

**Figure 1 materials-15-00122-f001:**
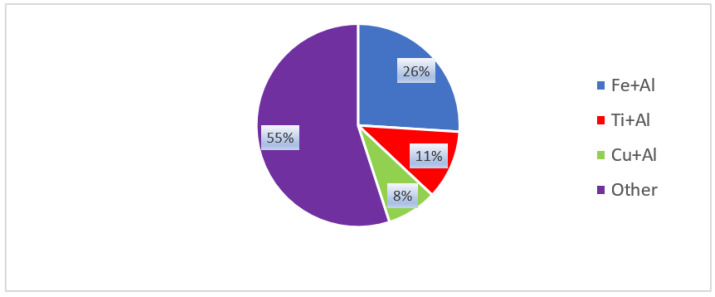
Distribution diagram of the topics of the articles on laser welding of dissimilar metals for 2016–2021 inclusively. Author’s own diagram.

**Figure 2 materials-15-00122-f002:**
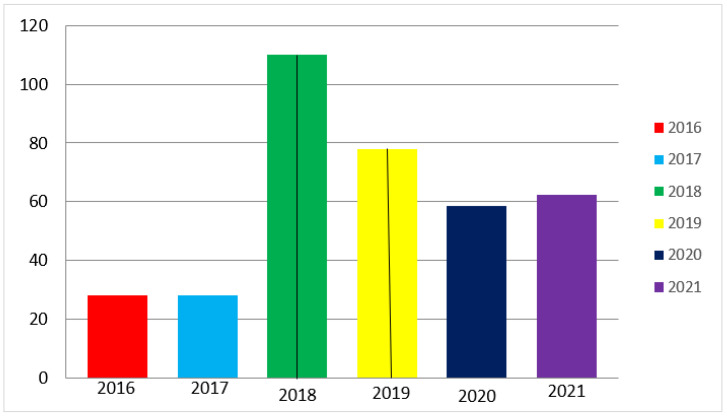
Distribution diagram of the number of articles on laser welding of dissimilar metals for 2016–2021 inclusively. Author’s own diagram.

**Figure 3 materials-15-00122-f003:**
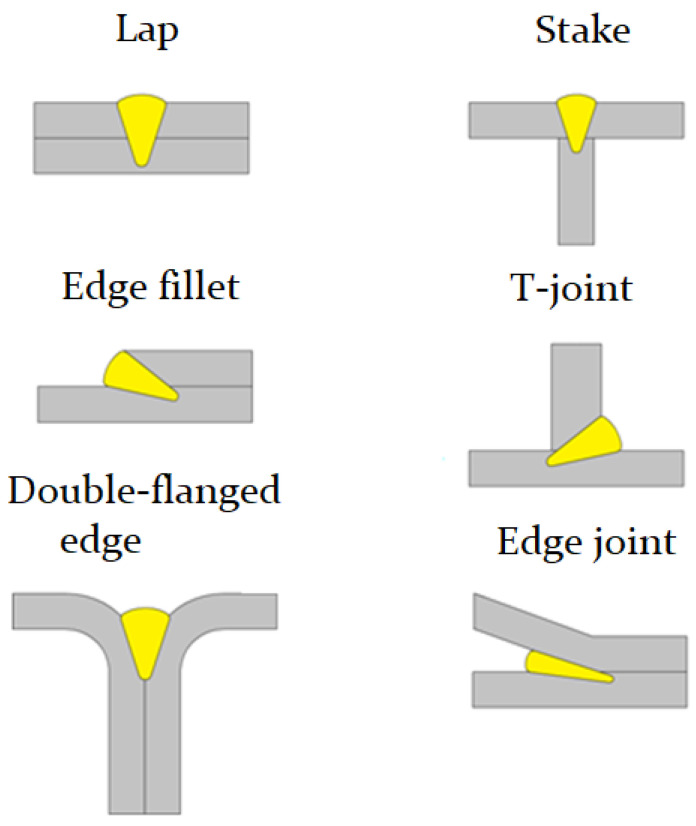
Joint configurations. Author own figure.

**Figure 4 materials-15-00122-f004:**
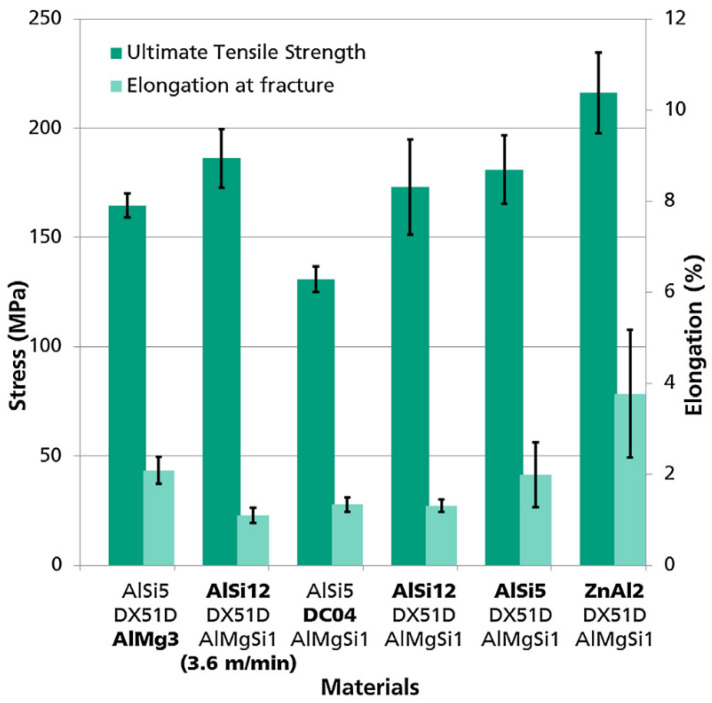
Tensile testing of double-flanged joints [25].

**Figure 5 materials-15-00122-f005:**
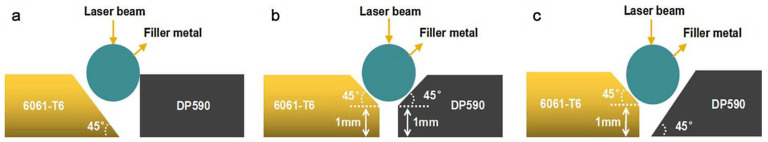
Diagram of weld cross section under different groove shapes: (**a**) square-shape groove at steel side, (**b**) half Y-shape groove at steel side, (**c**) half V-shape at steel side. Comparison of weld profile between experimental and numerical results: (**d**) experimental joint with square-shape groove, (**e**) experimental joint with half Y-shape groove, (**f**) experimental joint with half V-shape groove, (**g**) numerical joint with square shape groove, (**h**) numerical joint with half Y-shape groove, (**i**) numerical joint with half Y-shape groove [26].

**Figure 6 materials-15-00122-f006:**
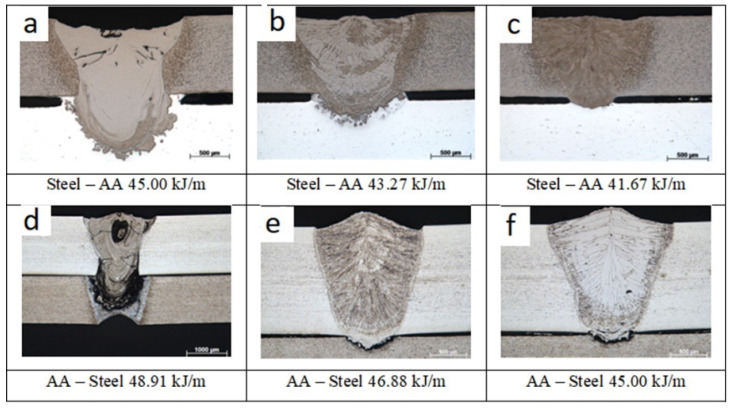
Microstructure of welded joints, (**a**–**c**)–impact of the beam from the steel side, (**d**–**f**)–impact of the beam from the aluminum side [2].

**Figure 7 materials-15-00122-f007:**
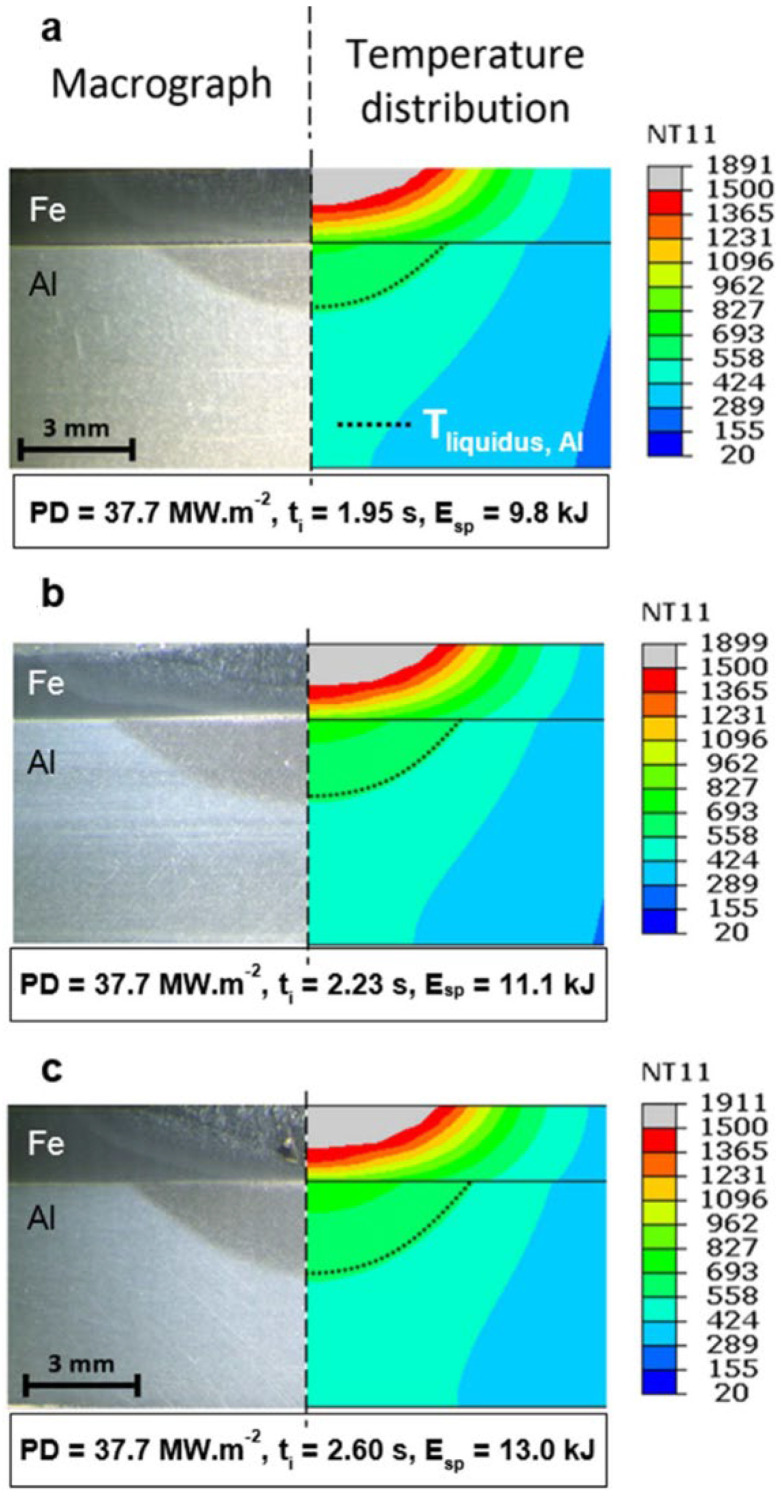
Comparison between the experimental and the FEM results: (**a**–**c**) macrograph vs. thermal profile. Reprinted with kind permission of Elsevier 2017 from Reference [28].

**Figure 8 materials-15-00122-f008:**
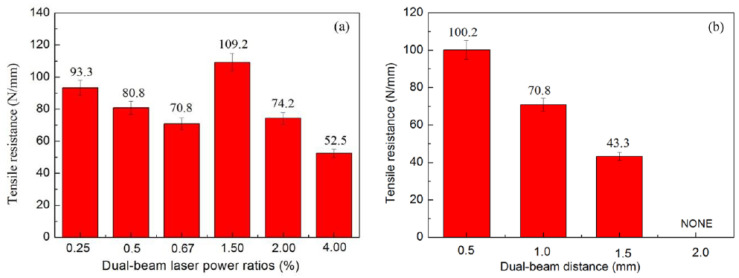
Effect of processing on tensile resistance of the steel/Al joints produced by dual-beam laser welding with side-by-side configuration: (**a**) dual-beam laser power ratios (%), (**b**) dual-beam distance (mm) [29].

**Figure 9 materials-15-00122-f009:**
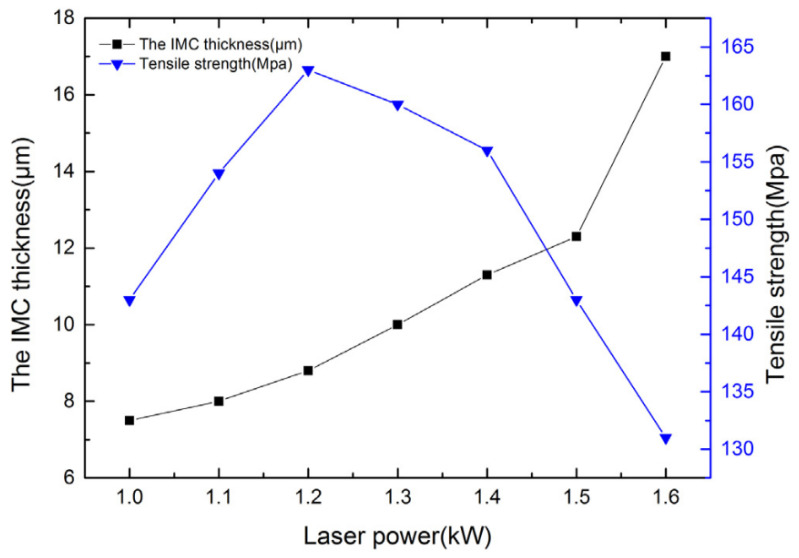
Effect of laser power on IMC thickness and tensile strength. Reprinted with kind permission of Elsevier 2017 from Reference [31].

**Figure 10 materials-15-00122-f010:**
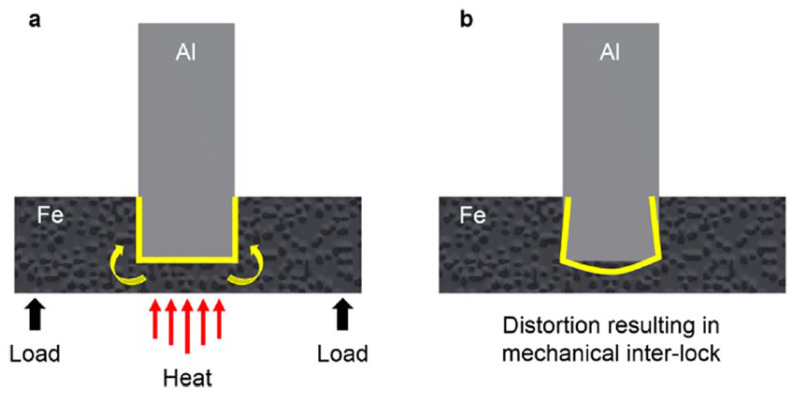
Schematic representation of the mechanical inter-lock induced during the welding process. (**a**) heating process, (**b**) mechanical interlock [32].

**Figure 11 materials-15-00122-f011:**
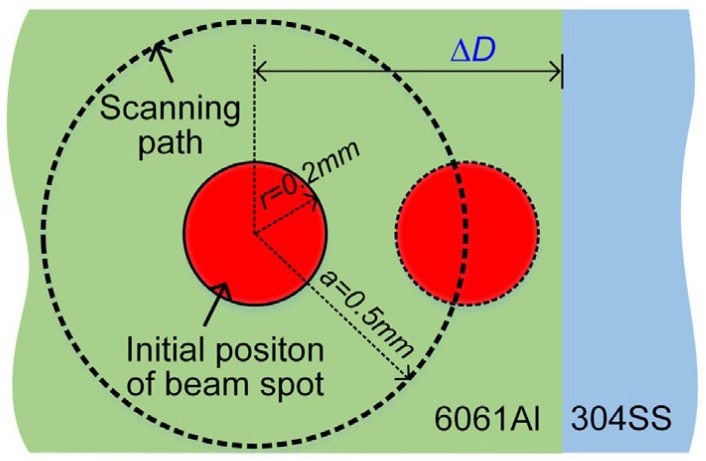
Diagram of offset and trajectory of the laser beam [33].

**Figure 12 materials-15-00122-f012:**
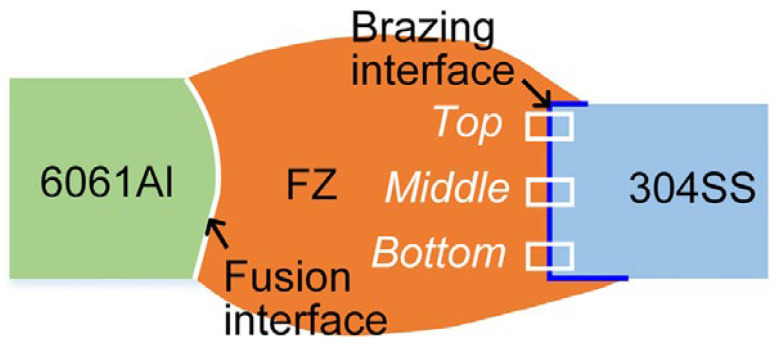
Schematic diagram of the macrostructure of the joint under study and a part of a detailed study of the microstructure (top, middle, bottom) [33].

**Figure 13 materials-15-00122-f013:**
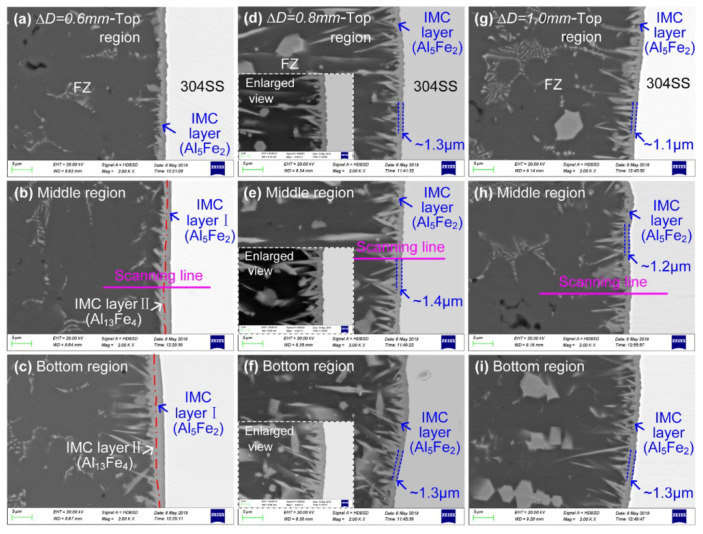
FZ/304SS interface obtained at different laser offset, (**a**–**c**) ΔD = 0.6 mm, (**d**–**f**) ΔD = 0.8 mm, (**g**–**i**) ΔD = 1.0 mm [33].

**Figure 14 materials-15-00122-f014:**
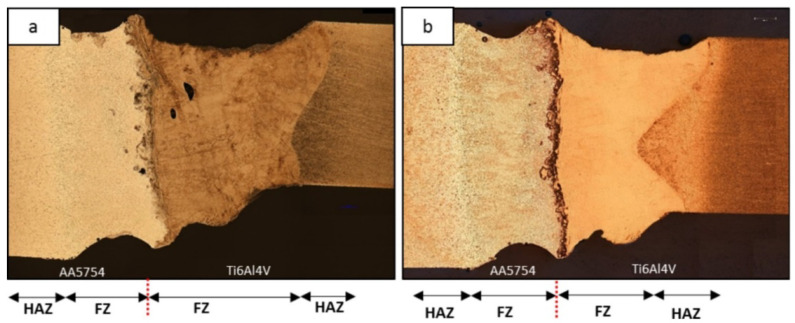
Macrostructure of the two joints: (**a**) 72 J/mm, (**b**) 36 J/mm [62].

**Figure 15 materials-15-00122-f015:**
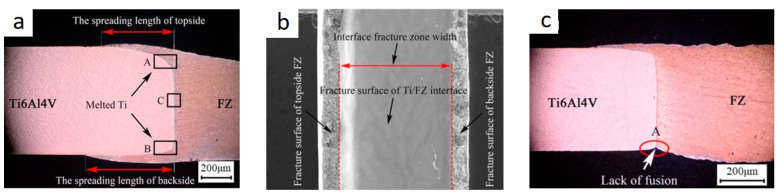
Macrostructure of a welded joint of the Ti and Al: (**a**) joint with a full top and bottom spreading, (**b**) fracture surface of a tensile speciment (**c**) joint with lack of fusion and bottom spreading [27].

**Figure 16 materials-15-00122-f016:**
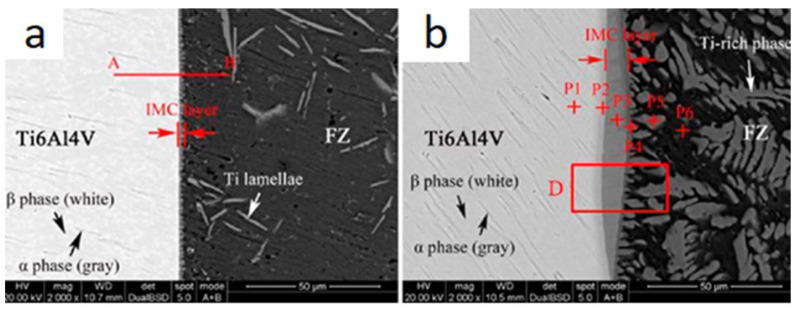
SEM image of the C (**a**) and A (**b**) region in Figure 15a [27].

**Figure 17 materials-15-00122-f017:**
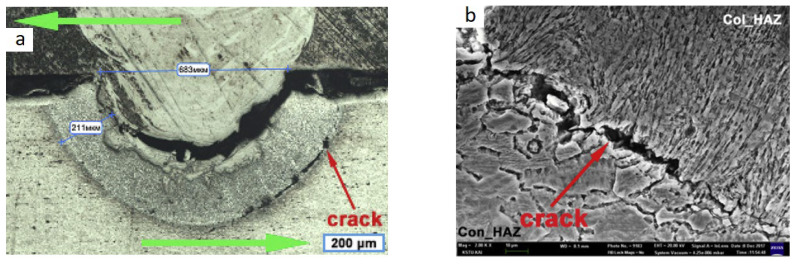
Microstructure of the Ti–Al after TSS (tensile shear strength): (**a**) welded joint microstructure (green arrows is a TSS direction), (**b**) crack microstructure [20].

**Figure 18 materials-15-00122-f018:**
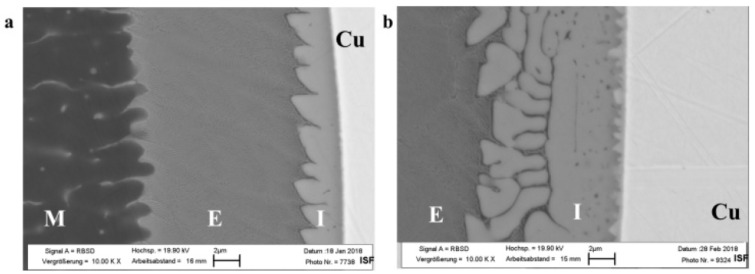
REM analysis (2 μm) of Al–Cu joint: (**a**) ∆x = 200 μm, (**b**) ∆x = 100 μm [68].

**Figure 19 materials-15-00122-f019:**
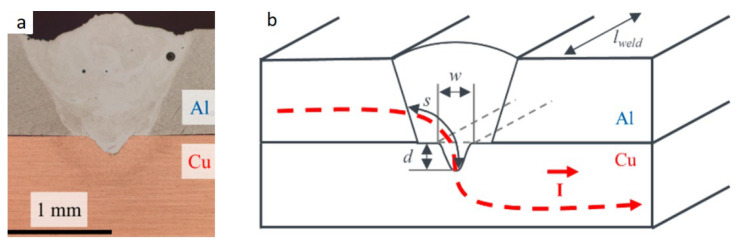
Welded joint of aluminum and copper: (**a**) macrostructure, (**b**) geometrical parameters of the weld seam [69].

**Table 1 materials-15-00122-t001:** Weldability of binary metal combinations [6].

Metal	Fe	Al	Ti	Ni	Mg	Cu
Fe	E	S	S	G	P	S
Al	S	E	S	S	S	S
Ti	S	S	E	S	P	S
Ni	G	S	S	E	S	E
Mg	S	S	S	S	E	S
Cu	S	S	S	E	S	E

E—Excellent, G—Good, S—Satisfactory, P—Poor, No Data.

**Table 2 materials-15-00122-t002:** Physical properties of Al, Fe, Ti, Cu [6].

Metal	Density (kg/m^3^)	Surface Tension (N/m)	Crystal Lattice Type/ Lattice Constant, nm	Atomic Radius, pm	Electronegativity (Pauling Scale)
Al	2700	0.914	FCC/0.4	143	1.61
Fe	7870	1.872	FCC, BCC/0.28	126	1.83
Ti	4500	1.650	HCP, BCC/0.29	147	1.54
Cu	8930	1.770	FCC/0.36	128	1.90

**Table 3 materials-15-00122-t003:** Thermophysical properties of Al, Fe, Ti, Cu [6].

Metal	Melting Point, °C	Coefficient of Thermal Conductivity, at 20 °C, W/(m × K)	Specific Heat Capacity (J/kg K)	Thermal Expansion Coefficient (10^6^ K)
Al	660	238	917	23.5
Fe	1539	78	456	12.1
Ti	1668	22	528	8.9
Cu	1083	397	386	17.0

**Table 4 materials-15-00122-t004:** Analysis of laser welding techniques for steel and aluminum, comparison of joint strength and IMC. Author’s own Table.

Alloys	Joint Configuration	Filler	Offset, mm	IMC Layer Thickness μm	Tensile, MPa, Shear Strength, N/mm^2^	Reference
DX51D + AlMgSi_1_ alloy	Lap	AlSi_5_, AlSi_12_ ZnAl_2_	-	10 μm	160 MPa 180 MPa 230 MPa	[25]
Steel 1.4301 + AA6016-T4	Lap	-	-	Not reported	3200 N	[2]
XF350 + 5083-H22	Lap	-	-	10–30 μm	500–600 N	[28]
Q235 + AA6061	Lap	-	-	8.4–23.7 μm	109.2 N	[29]
AISI 316 + AA5754	Butt	AISI 316	1.0	6 μm	Not reported	[30]
ST04Z steel + 5A06	Butt	Al80Zn8Mg7Mn2Si2	0.0	8.7 μm	163 MPa	[31]
6061-T6 + DP590 steel	Butt	AlSi_12_	to Al 0.4 mm	8.8 μm	108–145 MPa	[26]
AA6061-T6 + 304SS	Butt	ER4043	(∆D = 0.2; 0.4; 0.6; 0.8; 1.0 mm	1.3 μm	160 MPa	[33]
DH36 steel + AA5083	T-joint	-	-	5 μm	-	[32]

**Table 5 materials-15-00122-t005:** Value of the laser power and welding speed. Author’s own Table.

Welded Alloys	Joint Configuration/Thickness, mm	Laser Power, kW	Welding Speed, mm/s	Ref. No
Al/Fe				
DH36 + AA5083	Lap/2	5.5–6.5	5–10	[32]
DX51D + AlMgSi_1_	Lap/1	0.9–1.5	11–60	[25]
Q235 + AA6061	Lap/1.5	3.0	33	[29]
Steel 1.4301 + AA6061	Lap/2	3.75	-	[2]
XF350 + AA5083	Lap/1.5	4.46–5.57	5–6	[28]
DP600 + AA6061	Lap/2.5	3.5	8–10	[115]
DP1000 + AA1050	Lap/1	6.0–8.4	-	[50]
DP780 + AA5083	Lap/1	1.4–2.5	22	[103]
316SS + AA6063	Lap/0.2	0.070	150	[116]
22MnB5 + AA6061	Butt/1.5	2.1–3.6	8	[48]
304SS + AA6061	Butt/2	2.4	33	[33]
AISI316 + AA5754	Butt/2	2.5	33	[30]
DP590 + AA6061	Butt/2	2.2	18	[26]
5A06 + ST04Z	Butt/2	1.0–1.6	6–16	[31]
304SS + AA6061	Butt/2	1.8–3.3	33	[104]
304SS + AA6061	Butt/2	2.2–2.4	33	[105]
Al/Ti				
CP Ti + AA1050	Lap/0.3	1.0	170–830	[17]
CP Ti + AA2024	Lap/1	2.5–4.0	120–150	[20]
Ti6Al4V + AA6060	Lap/0.8	5.9–8.6	80	[109]
Ti6Al4V + 5A06	Lap/1.5	1.2	14	[113]
Ti6Al4V + AA2024	Lap/1.0	1.2	14	[102]
Ti6Al4V + 5A06	Butt/1.5	1–1.3	8–11	[27]
Ti6Al4V + AA6061	Butt/5	1.5–3.5	16	[63]
VT-20 + AA1461	Butt/2	3.3	66	[61]
Ti6Al4V + AA5754	Butt/2	1.2	16–32	[62]
Ti6Al4V + AA6061	Butt/3	0.5–1.5	10	[107]
TC4 + AA6082	Butt/1	3.2	3	[114]
Ti6Al4V + AA6061	Butt/2	1.5	40	[110]
Ti6Al4V + AA6061	Butt/2	2.0	6	[111]
Ti6Al4V + AA6061	Butt/1.6	1.0–1.5	16–50	[117]
Ti6Al4V + AA1050	Butt/1.2	1.6	7.5	[106]
Al/Cu				
Cu OF + AA1050	Lap/1	0.6	10	[67]
Cu OF + Al 99.5	Lap/1	1.5–3.25	100	[69]
CP Cu + CP Al	Lap/0.3	0.5–1.5	-	[118]
CP Cu + CP Al	Butt/2	2.0	10–35	[68]

## Data Availability

Data sharing is not applicable for this article.

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
