# Peer review of "A Review: Laser Welding of Dissimilar Materials (Al/Fe, Al/Ti, Al/Cu)—Methods and Techniques, Microstructure and Properties"

_materials, 2021, doi:10.3390/ma15010122_

Round 1

Reviewer 1 Report

This work carries out a quantitative and qualitative analysis (review) of articles for 2016 – 2021 on the topics of laser welding of dissimilar alloys, such as, Al/Fe, Al/Ti, and Al/Cu. However, the discussion in this article is not deep enough and some topics are not so important or directly related to this article, reviewer recommends that this paper need to be major revised before publication or reject. Detailed comments stated as follows:

  1. Laser welding of dissimilar alloy such as, the Al/Fe, Al/Ti, and Al/Cu has been discussed for the past decades. However, as described in the abstract, the source of the literature reviewed by the author of this article is from 2016 to 2021 (five years only) only. This may pose a question whether the contents and quality of the review article is good enough.
  2. We do believe, that the good articles or papers related to this review paper were published in 2020 or 2021, unfortunately, we would be able to see Quantitative analysis data of the world publications in Section 2 (Including Figure 1 and Figure 2)
  3. Authors should carefully check citations of the reference. We do find that some tables or figures do not cite the reference.
  4. Regarding to topics of laser welding of dissimilar alloys, Al/Fe, Al/Ti, and Al/Cu, authors should conduct an in-depth discussion on one topic at a time, Otherwise, the information provided in this article is limited to readers.

Author Response

Dear reviewers, thank you very much for yours precise review of my paper and for yours valuable comments which improve the quality of manuscript generally.

To avoid confusion, corrections in a text are highlighted for each reviewer with a separate color.

Reviewer 1

Reviewer 2

Reviewer 3

This work carries out a quantitative and qualitative analysis (review) of articles for 2016 – 2021 on the topics of laser welding of dissimilar alloys, such as, Al/Fe, Al/Ti, and Al/Cu. However, the discussion in this article is not deep enough and some topics are not so important or directly related to this article, reviewer recommends that this paper need to be major revised before publication or reject. Detailed comments stated as follows:

  1. Laser welding of dissimilar alloy such as, the Al/Fe, Al/Ti, and Al/Cu has been discussed for the past decades. However, as described in the abstract, the source of the literature reviewed by the author of this article is from 2016 to 2021 (five years only) only. This may pose a question whether the contents and quality of the review article is good enough.

Reply to Reviewers comments

The essence and peculiarity of this work are to analyze precisely the recent years in which laser welding and brazing are used very actively to join dissimilar alloys. For example, an analysis of 2021 papers made according to the WoS system showed that there were almost as many articles on the subject of Al/Ti laser welding as there were on Al/Fe laser welding. Whereas in previous years, this difference was almost three times greater in favor of Al/Fe. The revised version adds more information on articles of 2021.

  1. We do believe, that the good articles or papers related to this review paper were published in 2020 or 2021, unfortunately, we would be able to see Quantitative analysis data of the world publications in Section 2 (Including Figure 1 and Figure 2).

Reply to Reviewers comments

This remark was taken into account the Figure 1 and Figure 2 were corrected and supplemented. Over 25 references of 2020 – 2021 years were added.

  1. Authors should carefully check citations of the reference. We do find that some tables or figures do not cite the reference.

Reply to Reviewers comments

Thank you, dear reviewers, for such attentive work. In some tables (2, 3) and figures (1, 2, 3) that do not cite the reference, it is stated that this is "Authors own diagram or Author own figure".

  1. Regarding to topics of laser welding of dissimilar alloys, Al/Fe, Al/Ti, and Al/Cu, authors should conduct an in-depth discussion on one topic at a time, Otherwise, the information provided in this article is limited to readers.

Reply to Reviewers comments

The conclusions indicate the features for each pair Al/Fe, Al/Ti, and Al/Cu. And also, a detailed discussion of the analysis of 2021 papers, in which various interesting research results are found, was added. For example, the authors of [90] continued a more detailed study of the effect of wobbling parameters on Al/Fe butt joints of other authors, taking as a basis the experiment from [33]. The authors of [94] proposed using a titanium-aluminum compound in an aircraft wing structure, where titanium is used as a skin and is laser-welded to aluminum stringers and ribs. In [87], the results of the microstructure study, similar to my results, were obtained, and the simulation of the welding process of dissimilar alloys was carried out. About 25 more articles were analyzed. And thanks to the comment of the reviewer, it was found that many authors continue deeper studies of the results obtained earlier, putting forward hypotheses of the influence of laser welding parameters on the joint formation mechanisms. The above corrections were added in more detail to the Discussion and Conclusions section.

Reviewer 2 Report

 It was a pleasure to read this well-written review manuscript on the laser welding of dissimilar materials (Al/Fe, Al/Ti, 2 Al/Cu) – Methods and techniques, microstructure and properties. With the help many Tables and Figures, this manuscript analyzes clearly the influence of the basic techniques, methods, and means of laser welding of Al/Fe, Al/Ti, and 12 Al/Cu on the mechanical properties and thickness of the intermetallic compound.

Their quantitative analysis of research trends was supported by more than 80 scientific articles published in recent years. Key issues like the physical and thermos physical properties of Al, Fe, Ti, Cu were discussed. Their description of methods and techniques of laser welding of dissimilar metals, their analysis of laser welding techniques for steel and aluminum, comparison of joint strength and IMC were much detailed. Inclusion of laser welding of aluminum alloys to titanium alloys and laser welding of aluminum alloys to copper provide readers with a clear picture of the research achievements in this field. Important issue like the value of the laser power and welding speed was also provided.

Four important conclusions were finally drawn, by indicating promising technology like highly efficient laser welding of aluminum-based alloys and steels.

Based on the above mentioned facts, I would like to suggest the possible publication of the review manuscript by “Materials” and I also believe the considerable numbers of reads and citation this manuscript can bring to “Materials”.

Author Response

Dear reviewers, thank you very much for yours precise review of my paper and for yours valuable comments which improve the quality of manuscript generally.

To avoid confusion, corrections in a text are highlighted for each reviewer with a separate color.

Reviewer 1

Reviewer 2

Reviewer 3

It was a pleasure to read this well-written review manuscript on the laser welding of dissimilar materials (Al/Fe, Al/Ti, 2 Al/Cu) – Methods and techniques, microstructure and properties. With the help many Tables and Figures, this manuscript analyzes clearly the influence of the basic techniques, methods, and means of laser welding of Al/Fe, Al/Ti, and 12 Al/Cu on the mechanical properties and thickness of the intermetallic compound.

Their quantitative analysis of research trends was supported by more than 80 scientific articles published in recent years. Key issues like the physical and thermos physical properties of Al, Fe, Ti, Cu were discussed. Their description of methods and techniques of laser welding of dissimilar metals, their analysis of laser welding techniques for steel and aluminum, comparison of joint strength and IMC were much detailed. Inclusion of laser welding of aluminum alloys to titanium alloys and laser welding of aluminum alloys to copper provide readers with a clear picture of the research achievements in this field. Important issue like the value of the laser power and welding speed was also provided.

Four important conclusions were finally drawn, by indicating promising technology like highly efficient laser welding of aluminum-based alloys and steels.

Based on the above mentioned facts, I would like to suggest the possible publication of the review manuscript by “Materials” and I also believe the considerable numbers of reads and citation this manuscript can bring to “Materials”.

Reply to Reviewer comments

Dear reviewer, I suppose it's hard for you to imagine how pleasant it is to read such a good review of my work. In preparing and trying to publish this work, I spent a lot of effort. And it was worth it to get such pleasant comments from the reviewer at last. Thank you very much for your work and your wishes.

Reviewer 3 Report

The paper reviews the laser welding of dissimilar metals which is a very interesting topic both academically and industrially. The paper is well put together with the right mix of fundamentals and recent progress in the field. The paper can be published subject to the following minor revisions.

Specific comments:

  1. Where figures feature subsections, it is perhaps useful to refer them into subsections using a, b, c etc. For example, Fig. 3. Although this is a very minor observation as the figures are intelligible without this scheme as well.
  2. Table 4 presents shear strengths in N? Generally, shear strength is reported in Pa or N/m^2

  1. If the discussion can be improved by talking about the similarities between welded joints and laser powder bed fusion that can add further value. For example, printing metals such as copper and silver on steel plates have issues similar to what is discussed in this paper. A discussion in this regard can add value.

-Dissimilar metals deposition by directed energy based on powder-fed laser additive manufacturing. Journal of Manufacturing Processes Volume 43, Part A, 2019, Pages 83-97.

-Stable formation of powder bed laser fused 99.9% silver. Materials Today Communications, Volume 24, September 2020, 101195.

  1. Consider including a section on limitations and prospects regarding the welding of dissimilar metals.

Author Response

Dear reviewers, thank you very much for yours precise review of my paper and for yours valuable comments which improve the quality of manuscript generally.

To avoid confusion, corrections in a text are highlighted for each reviewer with a separate color.

Reviewer 1

Reviewer 2

Reviewer 3

The paper reviews the laser welding of dissimilar metals which is a very interesting topic both academically and industrially. The paper is well put together with the right mix of fundamentals and recent progress in the field. The paper can be published subject to the following minor revisions.

Specific comments:

  1. Where figures feature subsections, it is perhaps useful to refer them into subsections using a, b, c etc. For example, Fig. 3. Although this is a very minor observation as the figures are intelligible without this scheme as well.

Reply to Reviewers comments

Since there is only one reference to these individual figures, changes have been made in the text of the article. ……T-joint configuration (Fig. 3 “Lap”, “Stake”) has advantages…...

  1. Table 4 presents shear strengths in N? Generally, shear strength is reported in Pa or N/m^2

Reply to Reviewers comments

In all the above works, tensile shear strength is indicated in different ways. (N, N/mm, N/mm2). This remark was corrected.

  1. If the discussion can be improved by talking about the similarities between welded joints and laser powder bed fusion that can add further value. For example, printing metals such as copper and silver on steel plates have issues similar to what is discussed in this paper. A discussion in this regard can add value.

-Dissimilar metals deposition by directed energy based on powder-fed laser additive manufacturing. Journal of Manufacturing Processes Volume 43, Part A, 2019, Pages 83-97.

-Stable formation of powder bed laser fused 99.9% silver. Materials Today Communications, Volume 24, September 2020, 101195.

Reply to Reviewers comments

Indeed, the problems of a metallurgical nature in laser welding of dissimilar metals, described by the author, are similar to the problems arising from DMD or SLM technologies. Solutions to these problems can be borrowed.

The following text was added to the manuscript:

Obtaining permanent joints from dissimilar metals is an urgent challenge not only in the field of laser welding of dissimilar metals but also in such a promising and science-intensive manufacturing area as additive technologies (DMD – Direct Metal Deposition, SLM – Selective Laser Melting) [80, 81, 82, 83] and technology of recovery cladding [84, 85]. They also have problems with cracking, poor wettability, and flowability of the molten metal. As a rule, the substrate or the parts to be recovered differ in thermophysical and metallurgical properties from the metal being deposited. Therefore, the techniques and methods used in laser welding of dissimilar metals can be used in additive and cladding technologies.

  1. Consider including a section on limitations and prospects regarding the welding of dissimilar metals.

Reply to Reviewers comments

A part describing the latest achievements published in 2021 was added to the Discussion and Conclusions section. This part discusses the current prospects for the use of joints from dissimilar materials obtained by laser welding and the problems that researchers face.

Round 2

Reviewer 1 Report

Laser welding of dissimilar alloy such as, Al/Cu has been discussed for the past decades.  Literature reviewed by the authors of this article is from 2016 to 2021 (five years only) only. Reviewer may not be fully persuaded that it is enough to review relevant articles from 2016 to 2021. 

Some examples for Al/Cu laser welding before 2016 are listed as follows:

  1. Kah, P., et al., Factors influencing Al-Cu weld properties by intermetallic compound formation. International Journal of Mechanical and Materials Engineering, 2015. 10(1): p. 10.
  2. Zuo, D., et al., Intermediate layer characterization and fracture behavior of laser-welded copper/aluminum metal joints. Materials & Design, 2014. 58: p. 357-362.
  3. Hailat, M.M., et al., Laser micro-welding of aluminum and copper with and without tin foil alloy. Microsystem Technologies, 2012. 18(1): p. 103-112.

Authors should provide more information regarding to Al/Cu laser welding.  

Author Response

Reply to Reviewer comments:

The author is grateful to the Reviewer for such an attentive work. Correction of the remarks indicated both in the first and second reviews made it possible to improve the quality of the manuscript significantly.

The following text was added to the manuscript.

When analyzing publications, it was found that after laser welding, most researchers conduct mechanical tests (mainly tensile or shear strength tests) [71 – 77] while vital Al/Cu joints properties are thermal conductivity and electrical conductivity [78]. That is, the main area of application is the branch of electrical contact or heat transfer products and devices. Of course, some joints may be subjected to mechanical stresses, in particular cyclic ones with a small amplitude, the resistance against which can be ensured by mechanical fixation of parts welded from Al/Cu. In this case, it is necessary to pay great attention to the study of operational properties, in particular, electrical conductivity or degradation and changes in the IMC during electrical usage.

Especially it is necessary to highlight Reisgen U. et al. [79] investigating the operational properties of the Al/Cu welded joint when a current of 200 Amperes passes through it, which is a more in-depth continuation of the work describing the production of an Al/Cu welded joint [68]. Electrical usage was carried out for two weeks, after which the study did not find an increase in the IMC.

Another group of researchers led by Schmalen P. researched the robustness of the laser braze-welding process [80] and the measurement of the resistance of the Al/Cu overlap joint by the 4-wire method [81]. In this work [81], a circular laser beam wobbling was applied to increase the contact area of dissimilar metals, and the effect of laser power on electrical resistance was estimated, which is similar to the approach described in [69]. The optimal laser power is in the range of 900 – 1400 W.

A significant advantage of laser welding, which allows reducing the thickness of the IMC, is the use of high-speed laser welding up to 95 – 155 mm/s [75], 250 mm/s [71], 80 – 800 mm/s [76], which makes it possible to obtain a minimum IMC with a thickness of 1 – 4 μm with heterogeneous dual-phases (copper and γ-Al4Cu9 phases), which is also formed at solid-state welding [82]. The recently published work [71] on laser pressure welding is of particular interest, a process that combines laser welding with rolling welding, with a laser beam directed between two converging sheets of aluminum and copper. The authors of this work were the first to use this technique for welding the pure aluminum, then for pure copper, which are known to reflect laser irradiation strongly, and then for their combination. It should be noted that the use of longitudinal (relative to the welded plane of the sheets) laser beam wobbling, in this case, would significantly increase the area of the welded joint the tensile shear strength of the joint reached 111.94 MPa.

After tensile tests, failure areas of the Al/Cu welded joint were mainly observed in the copper and γ-Al4Cu9 [67, 71, 83] phase and the θ-CuAl2 phase [84]. In comparison with articles on laser welding of Al/Fe, Al/Ti, in articles on Al/Cu welding, little attention is paid to the hypothesis of the welded joint formation, in particular, the formation of various IMC phases and modeling of welding processes and their comparison with experimental data, which is an effective method of scientific research.

The main areas of research in the field of welding Al/Cu metals on the topic of which there are still not enough publications are surface modification before welding in order to increase wettability, the use of remote laser welding, which allows using ultra-high welding speeds, a more thorough study of the electrical conductivity of the obtained welded joints, as well as assessment of the economic efficiency of using the resulting welded joints. A serious challenge is the use of shorter wavelength lasers, the irradiation of which (1 – 2 kW) is difficult to focus at one spot to obtain maximum power density. In this case, it will be relevant to use laser beam wobbling systems, which allow heating the metal quickly and distributing the used small irradiation power evenly.

16 references were analyzed and added.

  1. Zhang JQ, Guo SH, Wang D, Xu JJ, Huang T, Xiao RS. Laser pressure welding of dissimilar aluminium and copper: microstructure and mechanical property. Science and Technology of Welding and Joining. 2021
  2. Yan SH, Shi Y. Influence of Ni interlayer on microstructure and mechanical properties of laser welded joint of Al/Cu bimetal. Journal of Manufacturing Processes. 2020;59:343-54.
  3. Liu HX, Jin H, Shao M, Tang H, Wang X. Investigation on Interface Morphology and Mechanical Properties of Three-Layer Laser Impact Welding of Cu/Al/Cu. Metallurgical and Materials Transactions a-Physical Metallurgy and Materials Science. 2019;50A(3):1273-82.
  4. Kermanidis AT, Christodoulou PI, Hontzopoulos E, Haidemenopoulos GN, Kamoutsi H, Zervaki AD. Mechanical performance of laser spot-welded joints in Al-Al/Cu solar thermal absorbers. Materials & Design. 2018;155:148-60.
  5. Zuo D, Hu SS, Shen JQ, Xue ZQ. Intermediate layer characterization and fracture behavior of laser-welded copper/aluminum metal joints. Materials & Design. 2014;58:357-62.
  6. Lee SJ, Nakamura H, Kawahito Y, Katayama S. Effect of welding speed on microstructural and mechanical properties of laser lap weld joints in dissimilar Al and Cu sheets. Science and Technology of Welding and Joining. 2014;19(2):111-8.
  7. Hailat MM, Mian A, Chaudhury ZA, Newaz G, Patwa R, Herfurth HJ. Laser micro-welding of aluminum and copper with and without tin foil alloy. Microsystem Technologies-Micro-and Nanosystems-Information Storage and Processing Systems. 2012;18(1):103-12.
  8. Shi WQ, Wang WH, Huang YL. Laser micro-welding of Cu-Al dissimilar metals. International Journal of Advanced Manufacturing Technology. 2016;85(1-4):185-9.
  9. Reisgen U, Olschok S, Holtum N. Influencing the electrical properties of laser beam vacuum-welded Cu-Al mixed joints. Journal of Laser Applications. 2019;31(2).
  10. Schmalen P, Plapper P, Cai WN. Process Robustness of Laser Braze-Welded Al/Cu Connectors. Sae International Journal of Alternative Powertrains. 2016;5(1):195-204.
  11. Schmalen P, Plapper P. Resistance Measurement of Laser Welded Dissimilar Al/Cu Joints. Journal of Laser Micro Nanoengineering. 2017;12(3):189-94.
  12. Kah P, Vimalraj C, Martikainen J, Suoranta R. Factors influencing Al-Cu weld properties by intermetallic compound formation. International Journal of Mechanical and Materials Engineering. 2015;10(1).
  13. Wu X, Zhang PL, Tang M, Li MC, Yu ZS, Li SW. Formation and Microstructure Characteristics in Spot Welding of Dissimilar Cu-Al Foil by Nanosecond Laser Scanning. Chinese Journal of Lasers-Zhongguo Jiguang. 2019;46(4).
  14. Zuo D, Hu SS, Shen JQ, Xue ZQ. Intermediate layer characterization and fracture behavior of laser-welded copper/aluminum metal joints. Materials & Design. 2014;58:357-62.

  1. Weigl M, Albert F, Schmidt M. Enhancing the Ductility of Laser-Welded Copper-Aluminum Connections by using Adapted Filler Materials. Lasers in Manufacturing 2011: Proceedings of the Sixth International Wlt Conference on Lasers in Manufacturing, Vol 12, Pt B. 2011;12:332-8.

  1. Hartel U, Ilin A, Bantel C, Gibmeier J, Michailov V, editors. Finite element modeling for the structural analysis of Al-Cu laser beam welding. 9th International Conference on Photonic Technologies (LANE); 2016 Sep 19-22; Furth, GERMANY 2016.
